# Th17/Treg Imbalance: Implications in Lung Inflammatory Diseases

**DOI:** 10.3390/ijms24054865

**Published:** 2023-03-02

**Authors:** Rony Thomas, Sai Qiao, Xi Yang

**Affiliations:** Department of Immunology, Faculty of Medicine, University of Manitoba, Winnipeg, MB R3E 0T5, Canada

**Keywords:** Th17 cells, Treg cells, inflammation

## Abstract

Regulatory T cells (Tregs) and T helper 17 cells (Th17) are two CD4^+^ T cell subsets with antagonist effects. Th17 cells promote inflammation, whereas Tregs are crucial in maintaining immune homeostasis. Recent studies suggest that Th17 cells and Treg cells are the foremost players in several inflammatory diseases. In this review, we explore the present knowledge on the role of Th17 cells and Treg cells, focusing on lung inflammatory diseases, such as chronic obstructive pulmonary disease (COPD), acute respiratory distress syndrome (ARDS), sarcoidosis, asthma, and pulmonary infectious diseases.

## 1. Introduction

The immune system acts as the guardian of the host and functions to defend against foreign antigens, induce self-tolerance, and promote immunological memory. However, it is not protective or beneficial all the time. The individual’s tissue components may be attacked by the immunological reaction resulting in autoimmune diseases in specific settings. It is certain that a single theory or mechanism cannot explain autoimmune diseases. As proposed by Shoenfeld and Isenberg, autoimmune diseases are caused by various factors, including immunological, genetic, hormonal, and environmental factors [1]. Non-genetic components rather than inherent components play a dominant role in determining disease susceptibility and severity, which has been demonstrated by the discordance of autoimmune diseases in identical twins [2]. Immunological factors play a vital role in the initiation, progression, and regression of autoimmune diseases. In a typical setting, T cells are tolerant to physiological levels of self-antigen. However, this state of tolerance breaks down in some individuals, resulting in autoimmune/inflammatory diseases. One of the critical features of inflammatory diseases is the deregulated Th1/Th17 responses, frequently accompanied by a reduction and/or alteration of regulatory T (Treg) cells. Th17 cells serve as inflammatory cells, which in excess, promote inflammatory diseases. On the other hand, Treg cells show suppressor function, which, when in failure, contributes to the same disease [3]. 

### 1.1. Th17 Cells

Initial studies by Infante-Duarte et al. identified CD4^+^ T cells producing IL-17A as a T helper cell subset distinct from Th1 and Th2 cell subsets [4]. This subset, called Th17 cells, predominantly produces interleukin-17A (IL-17A), IL-17F, IL-21, and IL-22 [5]. IL-17A, originally named CTLA8, was cloned and described by Rouvier et al. [6]. It is a homodimeric glycoprotein with 155 amino acids linked by disulfide bonds. IL-17F, also produced by Th17 cells, shows 55% similarity with IL-17A, and they form IL-17F homodimers, IL-17A homodimers, or IL-17A-IL-17F heterodimers [7]. IL-17 binds to its receptor (IL-17R), a transmembrane protein, highly expressed in rats and mice’s spleen, kidneys, liver, and lungs [8]. Th17 cells require the transcription factor, RORγt, and cytokine IL-6 in combination with transforming growth factor-β (TGF-β) for their differentiation [9]. IL-6 acts as a major factor guiding the differentiation of T cells into Th17 cells or Treg cells [9]. IL-21, together with TGF-β, also functions as an alternative pathway to generate Th17 cells [10]. Once they reach the site of inflammation, IL-17 released by Th17 cells stimulates the expression of pro-inflammatory cytokines like granulocyte-macrophage colony-stimulating factor, Granulocyte-colony stimulating factor, IL-6 and tumor necrosis factor-alpha (TNF-α) [11]. In addition, IL-17 also promotes the secretion of CXC chemokines, which attracts neutrophils in vivo [11]. Moreover, IL-17 stimulates the production of antimicrobial peptides, such as β-defensin and S100 proteins, providing defense against a wide range of microorganisms [12,13]. Furthermore, persistent secretion of IL-17 is involved in many inflammatory diseases [14,15,16,17]. Th17 cells also function as effective B cell helper cells by inducing B cell proliferation and antibody production [18]. 

### 1.2. Treg Cells

As the bias towards pro-inflammatory cytokines and cells induces the development and perpetuation of autoimmunity, immunoregulatory factors are thought to straighten out the laterality. Regulatory T cells are crucial members of the family of immunoregulatory cells that preserve self-tolerance and fine-tune the immune response. Treg cells suppress inflammation by cell-cell contact or releasing cytokines, such as IL-10 or TGF-β, and they require the transcription factor FoxP3 for their differentiation [3,19]. In recent years, research has identified two types of Treg cells called natural Treg cells (nTreg) and inducible Treg cells (iTreg). nTreg cells develop in the thymus, and when entering peripheral tissues, they suppress self-reactive T cells. Studies in both mice and humans found that nTreg cells constitute around 10% of CD4 T cells in the periphery [20]. They express FoxP3 before they are released from the thymus, and the expression of TGF-β helps in their maintenance of inhibitory function after they migrate from the thymus [3,19]. Inducible Treg cells develop from naive T cells in the secondary lymphoid organ upon antigen exposure. Following interaction with TCR, TGF-β induce the FoxP3 expression in CD4^+^ CD25^−^ cells, thereby, converting them to FoxP3^+^ CD4^+^ CD25^+^ cells [21]. These iTreg cells mediate their inhibitory activities by secretion of IL-10 or TGF-β, which is crucial for inhibiting overexuberant immune response [22] (Figure 1). 

## 2. Th17/Treg Cells in Lung Inflammatory Diseases

### 2.1. Chronic Obstructive Pulmonary Disease (COPD)

COPD is a chronic inflammatory lung disease characterized by airway and/or alveolar abnormalities that cause obstructed airflow from the lungs [23]. Studies over the last decade highlighted the relevance of maintaining the balance between Th17 cells and Treg cells to control the inflammatory response in COPD. An increased Th17 response is involved in the progression of Chronic Obstructive Pulmonary Disease (COPD) in both clinical and experimental studies [23]. Th17 cytokine, IL-17A, levels were higher in the sputum of patients with COPD stages 3 and 4 compared to non-smokers and healthy smokers [24]. Reduced numbers of Treg cells were observed in the bronchial epithelium of severe/very severe COPD patients than in those with mild and moderate COPD and healthy smokers [25]. Zheng et al. analyzed the Th17/Treg ratio in lung tissues of no-smoking and no-COPD (CS^−^COPD^−^), smoking and no-COPD (CS^+^COPD^−^), and COPD patients [26]. Flow cytometric analysis revealed a significantly higher Th17/Treg ratio in the COPD group compared to non-smoking patients [26]. 

In a mouse model of COPD induced by chronic cigarette smoke (CS) exposure for 4 and 24 weeks, mice chronically exposed to CS showed higher lung Th17 prevalence, increased retinoic acid orphan receptor (ROR)-γt mRNA, and Th17-related cytokines (IL-17A, IL-6, and IL-23) compared to control mice [27]. In contrast, Treg cell prevalence, Forkhead box (Fox)p3 mRNA, and Treg-related cytokine IL-10 were significantly reduced in mice chronically exposed to CS [27]. Similarly, the lungs of mice exposed to CS for 12 and 24 weeks also showed higher Th17 (CD4^+^IL-17^+^) cells, RORγt mRNA expression, and IL-6, IL-17, and TGF-β1 levels compared to the control group. In contrast, Treg cells, Foxp3, and IL-10 expression were reduced in the CS-exposed groups. Additionally, the frequencies of Tregs were negatively correlated with Th17 cells (33).

### 2.2. Acute Respiratory Distress Syndrome (ARDS)

Acute respiratory distress syndrome (ARDS) is an important cause of acute pulmonary failure with severe disease and mortality [28]. The most common cause of ARDS is bacterial or viral pneumonia [28]. ARDS is characterized by dysregulated inflammation, increased permeability of microvascular barriers, and uncontrolled activation of coagulation pathways [28]. Activation of several immune cells, including neutrophils, macrophages, and dendritic cells, plays an important role in the development of ARDS [29]. The involvement of CD4^+^ T cells has been revealed recently for the pathogenesis of ARDS. ARDS patients show a higher frequency of Th17 cells and IL-17 compared to the control group [29]. The Th17/Treg ratio is higher in the peripheral blood of ARDS patients compared with the healthy controls [29]. A higher Th17/Treg ratio is associated with more adverse outcomes in ARDS patients. Mechanistically, recent studies demonstrated that secreted phosphoprotein 1 (SPP1) exacerbated lung inflammation in ARDS by modulating Th17/Treg balance [30]. SPP1 reduces the ubiquitination and degradation of HIF-1α, which, in turn, leads to a higher Th1/Treg ratio. IL-33 production in LPS-induced ARDS is reported to increase the Th17/Treg ratio [31]. IL-33 deficiency inhibits the differentiation of T cells into Th17 cells and restores Th17/Treg balance. Consequently, IL-33 deficiency significantly reduces inflammation in LPS-induced ARDS, whereas recombinant IL-33 treatment exacerbates lung inflammation [31].

Modulation of Th17/Treg balance is relevant to resolve lung inflammation in acute lung injury (ALI), a milder form of ARDS [32]. Alanyl-glutamine (Ala-Gln) was administered to attenuate lung injury in a model of lipopolysaccharide (LPS)-induced ALI. Ala-Gln treatment increased the percentages of Tregs in the BAL fluid, whereas Th17 cells were suppressed, compared to the control group [32]. Similarly, losartan (an antagonist of angiotensin II type 1 receptor) treatment led to the inhibition of Th17 polarization after LPS-induced ALI [33]. These studies point out that the Th17/Treg imbalance is a potential indicator of the disease severity in ARDS patients.

### 2.3. Sarcoidosis

Sarcoidosis is an inflammatory disorder characterized by granulomatous inflammation that affects multiple organs, mostly the lungs and mediastinal lymph nodes [34]. Emerging studies suggest the pleiotropic functions of Th17 and Treg cells in the pathogenesis of sarcoidosis. Higher IL-17A cytokine production is observed in the BALF of patients with pulmonary sarcoidosis [34]. Moreover, a higher Th17/Treg ratio was observed in peripheral blood and BAL of patients with active and progressive sarcoidosis [35]. After treatment with corticosteroids, the level of Foxp3 mRNA was elevated in the peripheral blood, and expression of RORγt mRNA was reduced [35]. 

Moreover, Treg cells of the lungs of sarcoidosis patients exhibit a high level of inducible co-stimulator (ICOS) expression [36]. ICOS expression on Treg enhances the immune suppressive ability of Tregs [36]. In addition, recent studies suggest that the Treg/Th17 ratio can be used as a suitable biomarker for predicting sarcoidosis relapse along with other indicators [37]. The clinical characteristics of relapsed patients were compared with those of stable patients after corticosteroid withdrawal. In the relapsed patients, compared with the stable patients, Tregs cells were increased in parallel with an increase in Th17 cell. Nevertheless, after the retreatment of relapsed patients, Tregs were increased, leading to a higher Treg/Th17 ratio [37]. Tregs are found to accumulate at the sarcoidosis BAL, periphery, and peripheral blood of patients with active disease more than that of healthy controls [38]. Peripheral sarcoidosis Tregs showed an impaired ability to suppress effector CD4^+^ T cell proliferation [39]. Further studies on sarcoidosis patients with spontaneous clinical resolution showed that Treg cells regained suppressive ability in these patients [39]. Altogether, these studies imply that Th17/Tregs ratio and their functional capacity influence the progression or regression of pulmonary sarcoidosis.

### 2.4. Asthma

Asthma is a chronic inflammatory disease of the airways involving inflammatory cells such as mast cells, eosinophils, neutrophils, macrophages, and T lymphocytes [40]. Typically, asthmatic inflammation is mediated by excessively activated Th2 cells eosinophilia [40], but recent studies showed the involvement of cytokine IL-17A in multiple asthma pathogenesis, including neutrophilic inflammation, steroid insensitivity, activation of epithelial cells, and airway remodeling [41]. A large number of cells positive for IL-17 are reported in the sputum and bronchioalveolar fluids (BALFs) of asthmatic patients [42]. In addition, many reports identified that levels of IL-17A are correlated positively with the severity of asthma [43,44,45]. Inhibition of IL-17 in a model of LPS-induced asthma exacerbation aid in controlling Th2 and Th17 responses and signaling pathways associated with inflammation and remodeling [46].

Several studies highlight the relevance of maintaining Th17/Treg balance in controlling inflammation and the pathophysiology of asthma. In a chronic experimental model of asthma induction by administration of OVA, higher numbers of Treg cells as well as the release of IL-10 was observed with the efficacy of photobiomodulation (PBM) treatment [47]. PBM treatment also reduced recruitment of inflammatory cells, such as macrophages, neutrophils, and lymphocytes in the bronchoalveolar lavage fluid and release of cytokines into the BALFs [47]. Recent studies characterized the role of Treg cell subsets in the pathogenesis of allergic asthma [48]. The proportion of CD25^+^Foxp3^+^CD127^−^ Treg cells was reduced in the peripheral blood of allergic asthmatic patients compared to those of healthy subjects [48]. These circulating Treg cells in asthmatic patients expressed reduced CCR6 and IL-17 compared with healthy subjects. However, in a mouse model of allergic asthma induced by house dust mite (HDM), the CCR6^+^Treg cell number increased in the lung tissue [48]. Under the Th17 environment in the lung, CCR6^+^Treg cells differentiate into Th17-like cells. This Treg subset is the major pro-inflammatory Treg that promotes inflammation, producing IL-17 instead of immunoregulatory cytokines to exacerbate allergic asthma [48]. In children with allergic rhinitis accompanying bronchial asthma, a reduction in total Tregs was observed, whereas Th17 cells and plasma IL-17 levels were increased [49]. An imbalance of Th17/Treg was also correlated with airway hyperresponsiveness in asthmatic children [50]. In line with this, a combination of inhaled glucocorticoids (ICS) with long-acting β2-agonists (LABA) reduced Th17 cells and decreased the Th17/Treg ratio in house dust mites (HDM) allergic asthmatic children, leading to improvement clinically [51]. Moreover, the expansion of Th17 cells and reduction in regulatory CD4^+^ T cell subsets was identified as a mechanism by which leptin increases allergic asthma in obesity [52]. These studies using animal models and human studies support the relevance of maintaining Th17/Treg balance in controlling airway inflammation in asthma. 

### 2.5. Pulmonary Infectious Diseases 

In addition to their role in non-infectious inflammatory lung diseases, maintaining Th17 /Treg balance is important for protective immunity against lung infections. Human IL-17A and IL-17F are crucial for protective immunity against mucocutaneous candidiasis [53]. Treg cells prevent the differentiation of naïve T cells into Th17 cells and prevent the clearance of *Candida albicans* infection [54]. IL-17 is identified as a critical factor required for protective immunity to *Pneumocystis* infection. Administration of anti-IL-17 neutralizing antibody to wild-type mice infected with *P. carinii* resulted in severe fungal infection [55]. Similarly, regulatory T cells are recruited to the lung during the course of *Pneumocystis* infection in mice [56]. Depletion of the Treg population results in increased levels of IL-1β and IL-6, leading to increased lung injury [56]. Th17/Treg balance also acts as a critical factor for controlling lung inflammation during chlamydial infection. IL-17A produced by Th17 cells during chlamydial lung infection has a significant impact on the development of protective type 1 immunity [57,58,59]. Chlamydial lung infection of mice induced IL-17 production in lung and lymph nodes at earlier and later stages of infection [60]. Neutralization of IL-17 in mice resulted in higher body weight loss, bacterial burden, and more severe pathological changes in the lung compared with sham-treated control mice [60]. IL-17 neutralized mice exhibit reduced Th1 responses, increased Th2 responses, and altered DC phenotype. Moreover, the adoptive transfer of DC isolated from IL-17-neutralized mice failed to protect the recipients against challenge infection compared to DC from sham-treated mice [60]. Further examination identified IL-17A producers at early and later stages of chlamydial lung infection. Previous studies in Yang lab have shown that γδ T cells are the major producers of IL-17A at the initial stage of infection and quickly return to the background level at day 4 post-infection. Studies on the γδ T cell subsets further identified that Vγ4^+^T cells are the major IL-17A producing γδ T cell subset at the early stages of chlamydial lung infection [61]. IL-17A produced by γδ T cells has a promoting role in Th17 responses but no significant influence on T helper 1 response [57]. On the other hand, IL-17A produced by Th17 cells at later stages of chlamydial infection has a significant impact on the development of protective type 1 immunity [57]. These studies collectively suggest that IL-17A-mediated protection against chlamydial lung infection depends mainly on Th17 cells rather than γδ T cells [57]. In contrast to the findings in lung infections, either IL-17 receptor signaling or IL-23-dependent induction of IL-22 and IL-17 is reported to be indispensable for the resolution of genital tract infections [62,63]. 

On the other hand, higher Treg responses contribute to tissue pathology after chlamydial lung infection [58,59]. Treg cells are observed in the chlamydial infection site of both humans and mice [64,65,66]. Similarly, our recent studies suggested that NK cells provide protective immunity to chlamydial lung infection by maintaining Th17/Treg balance [67,68]. During *Chlamydophila pneumoniae (Cpn)* lung infection, NK cell depletion increased the number of Treg cells and IL-10-producing CD4^+^ T cells. The changes in T cell responses were associated with severe disease and bacterial load in the lung. Adoptive transfer of DCs from NK cell-deficient mice induced Treg cells in the recipient mice, which promotes pathological response [67]. In the mice model of *Chlamydia muridarum* lung infection, NK cell depletion resulted in lower IL-17 cytokine production and Th17 cells [68]. On the other hand, NK cell-depleted mice exhibited increased production of CD4^+^ CD25^+^ Foxp3^+^ T cells resulting in a reduced Th17/Treg ratio [68]. These studies highlight that an imbalance between Treg cells and Th17 cells acts as a major factor determining the severity of lung infection.

In some lung infections, a higher Th17/Treg ratio contributes to the severity of the disease. An imbalance in Th17/Treg ratio was also observed in patients with *Mycoplasma pneumoniae* (MP) infection [69]. Refractory MP pneumonia patients showed a higher Th17/Treg ratio than non-refractory MP pneumonia patients and the control group [69]. It suggests that patients with refractory MP pneumonia have a balance shifted toward the induction of inflammatory responses, while patients with non-refractory MP pneumonia have a balance shifted toward the inhibition of inflammatory responses. It is found that Respiratory syncytial virus (RSV) infection, a major causative agent of pneumonia in infants, increases Th17/Treg ratio, thereby disrupting asthmatic tolerance [70]. Similarly, studies by Qin et al. showed that infection of human bronchial epithelial cells by RSV induces differentiation of lymphocytes into Th17 cells while inhibiting differentiation into Treg cells [71]. In the peripheral blood of infants with RSV bronchiolitis, the percentage of Tregs and the levels of IL-10 and TGF-β were significantly lower compared to those with non-RSV pneumonia and healthy infants [72]. On the other hand, the percentage of Th17 cells and the level of IL-17 were significantly higher in infants with RSV bronchiolitis compared to those with non-RSV pneumonia and healthy infants [72]. 

## 3. Therapeutic Strategies Targeting Th17/Treg Cells in Lung Inflammatory Diseases

Based on the previous studies, it is clear that maintaining Th17/Treg balance by blocking Th17 cell differentiation or inducing Treg activation will effectively treat various inflammatory diseases. To control inflammation in COPD, PBMC from COPD patients was treated with Tiotropium (anticholinergic drug) and Olodaterol (long-acting β2-agonist) [73]. The treatments reduced the percentage of T cells co-expressing acetylcholine (Ach)IL-17A, AChIL-22, and AChRORγt while increasing the Foxp3-expressing T cells in PBMC from COPD patients [73]. In a mice model of lung inflammation and emphysema induced by elastin peptides (EP) intranasally, treatment with erythromycin reduced the Th17 cells while increasing the Treg response [74]. In addition, erythromycin treatment suppressed cigarette smoke extract-exposed dendritic cell-mediated polarization of CD4^+^ T cells into Th17 cells [75]. The effect of treatment with N-Acetylcysteine (NAC) was tested in COPD patients [76]. Oral administration of NAC significantly reduced the frequencies of Th17 cells in the peripheral blood of the COPD patients group compared to those in the control group [76]. On the contrary, Treg cell frequencies increased in treated COPD patients. Mechanistically, NAC regulated Th17/Treg balance by downregulating HIF1-α, which induced RORγt transcription and Foxp3 protein degradation [76]. Similarly, simvastatin, a clinically used cholesterol-lowering agent, inhibits IL-17 but enhances IL-10 to reverse Th17/Treg imbalance in COPD patients [77]. 

Blocking IL-17 or cytokines that activate Th17 cells is used as a strategy to alleviate asthmatic inflammation. Anti-IL-17 monoclonal antibodies (mAb) treatment before allergen inhalation strongly reduced bronchial neutrophil infiltration in a mouse model of allergic asthma [78]. Resolvin E1 (RvE1) suppresses IL-23 and IL-6 production to promote the resolution of allergic airway inflammation [79]. IL-23 and IL-6 cytokines, in turn, promote the survival and differentiation of Th17 cells [79]. Dopamine D1-like receptor antagonist attenuates allergic airway inflammation by inhibiting the production of IL-17 and infiltration of Th17 cells in the lung [80]. Similarly, RAPA (inhibitor of mammalian target of rapamycin) treatment inhibits OVA-induced neutrophilic airway inflammation by suppressing Th17 cell differentiation [81]. Administration of rosiglitazone or pioglitazone (peroxisome proliferator-activated receptor agonists) reduced infiltration of inflammatory cells and inhibited IL-17 expression after OVA inhalation [82]. On the other hand, rosiglitazone or pioglitazone administration further enhanced IL-10 cytokine in lung tissues after ovalbumin inhalation [83]. Recent studies demonstrated that Tregitopes (T regulatory epitopes) attenuated airway hyperresponsiveness and inflammation in a murine model of allergic asthma [84]. Tregitopes induce highly suppressive allergen-specific Tregs, which inhibit inflammatory response [84]. Antigen-specific immunotherapy (ASIT) is one of the major methods of in vivo induction of Tregs in allergic asthma. Grass tablet sublingual immunotherapy downregulates the Th2 cytokine response and increases regulatory T-cells [85]. Similarly, dual sublingual immunotherapy enhanced regulatory T cell function with lower DNA methylation of CpG sites within the Foxp3 locus [86]. In a mouse model of allergic asthma, exogenous Semaphorin 3E (Sema3E) treatment reduces Th17 cytokine response leading to diminished collagen deposition, airway hyperresponsiveness, and eosinophilic inflammation [87]. The therapeutic function of Sema3E is also investigated in bacterial infection [59]. Exogenous Sema3E treatment protects mice from chlamydial infection with reduced chlamydial load, lower body weight loss, and pathological changes in the lungs [59]. Sema3E treatment resulted in higher Th1/Th17 response but reduced Treg response in the lungs of chlamydia-infected mice compared to saline-Fc treated mice [59]. 

## 4. Conclusions

The importance of the balance between pro-inflammatory and anti-inflammatory cytokines and cells in maintaining immune homeostasis is widely acknowledged. Th17 cells promote inflammation and pathology, whereas Treg cells maintain self-tolerance. The balance between inflammation and self-tolerance is disrupted, leading to inflammation. Developing therapeutic approaches focusing on Th17/Treg imbalance is likely an efficient way to prevent and/or treat various inflammatory diseases.

## Figures and Tables

**Figure 1 ijms-24-04865-f001:**
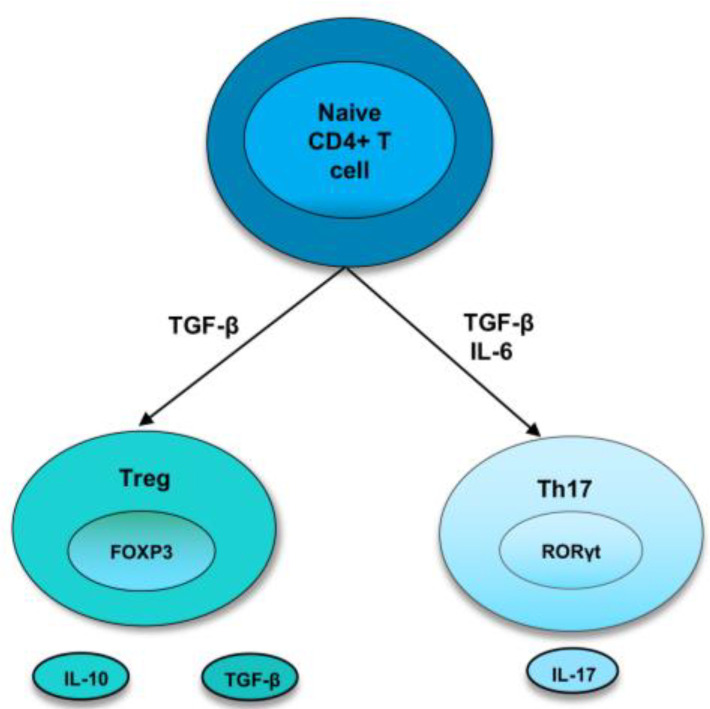
Differentiation of naive T cells into Th17 and Treg cells. In naive CD4^+^ T cells, TGF-β induce the development of Tregs by promoting Foxp3 expression. Treg cells express cytokines, IL-10 and TGF-β. However, in the presence of IL-6 and TGFβ, RORγt is induced, leading to a Th17 phenotype.

## Data Availability

Data is contained within the article.

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
