# Peer review of "Th17/Treg Imbalance: Implications in Lung Inflammatory Diseases"

_ijms, 2023, doi:10.3390/ijms24054865_

Round 1

Reviewer 1 Report

Summary: In this review, the authors Thomas et al., investigated the current knowledge on the role of Th17 10 cells and Treg cells in inflammatory diseases emphasizing on lung inflammatory diseases, multiple sclerosis (MS), rheumatoid arthritis (RA), systemic lupus erythematosus (SLE), Sjogren’s syndrome (SS), and inflammatory bowel disease (IBD) .The authors stated that, an imbalance of Th17/Treg was also correlated with airway hyperresponsiveness in asthmatic children .They highlighted the importance of Th17/Treg balance in relationship with the inflammatory diseases. The authors concludes that maintaining Th17/Treg balance by blocking Th17 cell differentiation or inducing Treg activation will effectively treat various inflammatory diseases.

Comments:The authors clearly included sufficient information supporting their conclusion that pro-inflammatory and anti-inflammatory cytokines and cells in maintaining immune homeostasis are highly recognized.

Author Response

Thank you so much for your valuable comments. As suggested by the Academic Editor, we have revised the manuscript to focus more on lung inflammatory diseases.

Reviewer 2 Report

This review will be an important contribution towards therapeutic interventions involving Th17/Tregs in various autoimmune disorders. I have a few specific suggestion for the authors: 

- It will be better to it as a table in 3, Therapeutic implications for targeting Th17/Treg cells. 

- Authors should cite some pioneer papers from Vijay K Kuchroo group. Same for Tregs from Rudensky group.

- 1.1 and 1.2, A figure showing the differentiation signals for Th17 and Treg would be ideal. 

Author Response

Thank you so much for your valuable suggestions. Based on the instruction of the academic editor and the special issue's scope, we have significantly revised the manuscript to focus on maintaining Th17/Treg balance in lung inflammatory diseases. Therefore, the figure showing therapeutic strategies for maintaining the balance between Th17/Treg cells in autoimmune diseases was removed. Based on your valuable suggestion, we have cited several relevant references from the Rudensky group for Treg (Section 1.2) and Vijay Kuchroo group (Section 1.1). Also, a figure showing differentiation signals for Th17 and Treg is provided as Figure 1.

Round 2

Reviewer 2 Report

Authors have included the recommended changes and the manuscript should be accepted in the current form.